# Radiomics and Artificial Intelligence for Outcome Prediction in Multiple Myeloma Patients Undergoing Autologous Transplantation: A Feasibility Study with CT Data

**DOI:** 10.3390/diagnostics11101759

**Published:** 2021-09-24

**Authors:** Daniela Schenone, Alida Dominietto, Cristina Campi, Francesco Frassoni, Michele Cea, Sara Aquino, Emanuele Angelucci, Federica Rossi, Lorenzo Torri, Bianca Bignotti, Alberto Stefano Tagliafico, Michele Piana

**Affiliations:** 1LISCOMP, Dipartimento di Matematica, Università di Genova, Via Dodecaneso 35, 16146 Genova, Italy; schenone@dima.unige.it (D.S.); campi@dima.unige.it (C.C.); francesco.frassoni19@gmail.com (F.F.); 2Ospedale Policlinico San Martino-IRCCS, Largo Rossana Benzi 10, 16132 Genova, Italy; alida.dominietto@hsanmartino.it (A.D.); michele.cea@unige.it (M.C.); sara.aquino@hsanmartino.it (S.A.); emanuele.angelucci@hsanmartino.it (E.A.); lorenzo.torri@gmail.com (L.T.); bianca.bignotti@hsanmartino.it (B.B.); 3Dipartimento di Medicina Interna, Università di Genova, Viale Benedetto XV, 16132 Genova, Italy; 4Dipartimento di Medicina Sperimentale, Università di Genova, Via L. B. Alberti 1, 16132 Genova, Italy; federica.rossi@unige.it; 5Dipartimento di Scienze della Salute, Università di Genova, Via Pastore 1, 16132 Genova, Italy; 6CNR-SPIN Genova, Via Dodecaneso 33, 16146 Genova, Italy

**Keywords:** multiple myeloma, computerized tomography, image processing, pattern recognition, artificial intelligence

## Abstract

Multiple myeloma is a plasma cell dyscrasia characterized by focal and non-focal bone lesions. Radiomic techniques extract morphological information from computerized tomography images and exploit them for stratification and risk prediction purposes. However, few papers so far have applied radiomics to multiple myeloma. A retrospective study approved by the institutional review board: *n* = 51 transplanted patients and *n* = 33 (64%) with focal lesion analyzed via an open-source toolbox that extracted 109 radiomics features. We also applied a dedicated tool for computing 24 features describing the whole skeleton asset. The redundancy reduction was realized via correlation and principal component analysis. Fuzzy clustering (FC) and Hough transform filtering (HTF) allowed for patient stratification, with effectiveness assessed by four skill scores. The highest sensitivity and critical success index (CSI) were obtained representing each patient, with 17 focal features selected via correlation with the 24 features describing the overall skeletal asset. These scores were higher than the ones associated with a standard cytogenetic classification. The Mann–Whitney U-test showed that three among the 17 imaging descriptors passed the null hypothesis. This AI-based interpretation of radiomics features stratified relapsed and non-relapsed MM patients, showing some potentiality for the determination of the prognostic image-based biomarkers in disease follow-up.

## 1. Introduction

Plasma cell dyscrasias (PCDs) include monoclonal gammopathy of undetermined significance (MGUS), smoldering multiple myeloma (SMM), and full-blown multiple myeloma (MM) [1]. Around 5% of the population over 70 are MGUS patients, and for around 1% of them MGUS will probably turn into MM every year. Around 10% of the SMM population evolves into full-blown MM, whose early mortality is nowadays around 28% five years after diagnosis [2]. MM is still an incurable disease, whose definition relies on the International Myeloma Working Group (IMWG) consensus updates, which is characterized by a notable clinical heterogeneity, so that the search for consolidated biomarkers predicting the disease outcome and progression is still a crucial open issue [3,4,5].

The presence of either single or multiple bone lesions is a typical signature of MM, which is related to the proliferation of tumor cells from a single clone, so that the unbalanced activation of osteoclasts erodes the medullary and even the cortical bone [6].Therefore, the CRAB criteria of IMWG underlines the importance of imaging for MM assessment, and recent staging systems rely on the use of imaging modalities like magnetic resonance imaging (MRI), computerized tomography (CT) and hybrid positron emission tomography with CT (PET/CT) [3,4,6,7,8,9,10,11,12,13,14,15]. However, just the availability of different imaging modalities and the high variability of image interpretation imply a notable heterogeneity as far as the use of imaging for MM clinical practice is concerned [6,12,16].

At a more specific level, the limitations of the use of imaging for MM assessment are essentially due to three open issues: the lack of accuracy in differentiating focal from diffuse patterns, the difficulty in extracting reliable prognostic biomarkers from pattern allocation, and the low agreement in staging MM patients based on imaging outcomes [17,18].

The application of pattern recognition algorithms for the extraction of radiomics descriptors from images of MM patients and the post-processing of such radiomics features by means of procedures based on artificial intelligence (AI) are nowadays introducing a novel approach for increasing the reliability of imaging in MM clinical assessment [17,18,19,20]. The objective of the present study is to assess the feasibility of an AI-based approach for the automatic stratification of MM patients from CT data, and for the automatic identification of radiological biomarkers with a possible prognostic value. Specifically, relying on radiomics and AI-based computational analysis [19,21,22], this feasibility study shows that a set of descriptors of the focal lesions in MM X-ray CT at diagnosis allows for the automatic stratification of a cohort of MM patients who have undergone transplantation in two clusters, whose characteristics can be interpreted via comparison with clinical data, biological biomarkers, and the clinical outcome of the disease.

## 2. Materials and Methods

### 2.1. Study Populations, Inclusion Criteria, and Risk Stratification

This study was performed according to the Declaration of Helsinki and the International Conference on Harmonization of Good Clinical Practice Guidelines. An institutional review board was obtained (054REG2019). All patients signed informed consent for retrospective research before CT examination; data collection did not influence patient care. We considered 51 consecutive patients (mean age, 56 years ± 8; range, 31–73 years; 18 females; 33 males) admitted to the Hospital (BLIND for REVIEW) in the last five years because of biopsy confirmed MM. Inclusion criteria were baseline whole-body CT from the Hospital PACS or outpatient clinic. Among these 51 patients, we selected the 33 presenting at least one focal lesion in one of the CT slices, i.e., at least one >5 mm lytic lesion in the axial or extra-axial skeleton [17,18] (see Figure 1). Two radiologists blinded to the diagnosis and to each other’s conclusion assessed whether the CT pattern was diffuse or focal, and, for each patient presenting at least one focal lesion, we identified the largest one.

Risk stratification was performed at diagnosis by the Revised International Staging System (ISS) combining serum beta2-microglobulin and serum albumin, lactate dehy-drogenase for three-stage classification, and cytogenetics determining a binary normal-high risk stadiation [23,24]. Table 1 provides a summary of the clinical features (diameter of focal lesion: mean: 19.9 mm, STD: 13.4 mm, min: 4.5 mm, max: 62.4 mm).

### 2.2. Image Analysis

To compute each patient’s overall skeletal asset, we utilized a published software tool (Bone-GUI, http://mida.dima.unige.it/software/bone-gui/; accessed on 20 September 2021) [25] combining thresholding and active contours. For each subject, Bone-GUI provided 24 features. Separately for the whole, axial, and skeleton districts, it computed the following: the mean medullary Hounsfield value with standard deviation, the volume of the global medullary asset, the mean cortical Hounsfield value with standard deviation, the volume of the cortical asset, the rate of volume occupied by the medullary tissue, and the overall volume.

We also applied an open source tool for radiomics (Slicer, https://www.radiomics.io/slicerradiomics.html; accessed on 20 September 2021) [26,27,28] to the 33 lytic lesions on the compact bone tissue to extract 109 Slicer features for each focal lesion.

### 2.3. Reduction of Redundancy

Our AI-based analysis for patients’ stratification utilized Slicer features as the input. To reduce information redundancy, we considered two approaches. In the first approach, principal component analysis (PCA) [29] projected the feature space onto a principal components’ subspace explaining at least 80% of the data variance. In the second approach, we performed two Pearson’s correlation processes (*p* > 95%) involving the Slicer features and (a) the binary feature encoding patient’s relapse one year after transplantation, and (b) all 24 Bone-GUI features. We applied PCA to the features selected using the two correlation processes. Figure 2 illustrates this redundancy reduction pipeline.

### 2.4. Clustering

Clustering organized a set of unlabeled samples into clusters based on data similarity [30]. Data partition was obtained by minimizing a cost function involving the distances between the data and cluster prototypes. In Fuzzy C-Means (FCM) a degree of membership is assigned to each sample with respect to each cluster. In addition to FCM, we applied a non-linear approach based on the filtering of an extended version of the Hough transform (HTF) [31], according to the following steps (Figure 3):Downstream of the PCA process, the two-dimensional feature space given by the two components explaining most of the data variance (namely, PC1 and PC2) was constructed for each data set.Given a feature space, the Hough transform of each point in the patient’s set with respect to the family of all parabolas was computed. As this family was characterized by three parameters, i.e., its equation is y_PC2 = ax_PC1^2 + bx_PC1 + c, with a, b, and c being the parameters, and the corresponding parameter space has three dimensions.The Hough accumulator was computed by counting the number of times each Hough transform passed through one of the cells of the discretized parameter space.The Hough accumulator was filtered by a 5-pixel-side cube centered on the pixel with a maximum grey value. This cube was the smallest one enclosing the cells, with accumulator values higher than 50% of the maximum [32].

Each line passing through the filtered region was projected back to the image space, thus generating a cluster of points in a strip around the parabola corresponding to the maximum in the Hough accumulator. The remaining points represent the second cluster made of points outside of the strip of parabolas previously identified.

## 3. Results

### 3.1. Clinical Findings

Focal lesion searching led to the selection of 33/51 (65%) patients (mean age, 56 years ± 7; range, 45–69 years; 12 females; 21 males) whose imaging data were considered for our computational analysis. Inter-observer agreement in differentiating diffuse from focal pattern between the two groups of radiologists resulted in 0.75 (95% Confidence Interval: 0.31–0.67) and 0.96 (95% Confidence Interval: 0.79–0.99) for the selection of patients with focal lesions.

### 3.2. AI-Based Analysis

The AI-based analysis involved three data sets (see Table 2): data set 1, made of all 109 local features extracted by Slicer from each focal lesion; data set 2, made of the eight local features mostly correlating with the relapsed/non-relapsed binary feature; and data set 3, made of the 17 local features mostly correlating with the 24 Bone-GUI global features. The application of PCA to these three data sets led to three features spaces, with *n* = 5 axes for data set 1, *n* = 3 axes for data set 2, and *n* = 2 axes for data set 3.

In each one of these three feature spaces, FCM and HTF computed two clusters: in each cluster, the black circles are associated with patients that underwent relapse within one year of bone marrow transplantation. Cluster A (B) contained the maximum (minimum) number of relapsed patients; in Figure 4, Clusters A (B) are coded with blue (orange). Table 3 contains a summary of how the clusters are populated for each of the three data sets and each of the two AI methods utilized for the analysis.

In order to assess the performances of the clustering algorithms, we computed the confusion matrices for the observed relapsed patients; specifically, we counted the number of true positives (TPs), true negatives (TNs), false positives (FPs), and false negatives (FNs) using cluster A as the reference cluster for the “relapsed” class and cluster B as the reference cluster for the “non-relapsed” class. Using the entries of such matrices, we computed four different skill scores:

Sensitivity = TP/(TP + FN)

Specificity = TN/(TN + FP)

Youden’s index = Sensitivity + Specificity − 1

Critical Success Index (CSI) = TP/(TP + FN + FP).

We show that the CSI ranged from 0 to 1 and it was higher as much as the number of FPs and FNs was small, regardless the number of TNs. CSI is therefore a useful score in conditions like the one we considered here, where we had an unbalanced data set with more non-relapsed cases than relapsed ones.

We tested the robustness of our results by performing a bootstrap analysis on the set 33 17-dimension feature vectors of that set. We constructed 100 random realizations of training sets made of 20 feature vectors (of which 10 representing relapsed patients) and, for each realization, we applied the HTF clustering process. Then, for each realization of the training set, we computed the membership cluster for each one of the remaining 13 vectors representing the test set. Repeating this procedure for each one of the 100 realizations of the training-test set pairs led to the construction of 100 confusion matrices and, therefore, to 100 sets of skill score values that we averaged in Table 4, together with the corresponding standard deviations. We also performed a bootstrap analysis on the cytogenetics values. In order to compute the entries of these last confusion matrices, we compared the relapse/non-relapse with the high/standard cytogenetic stages: a relapsed patient with a “high” cytogenetic stage was a TP event, while a relapsed patient with a “standard” cytogenetic stage was an FN. A non-relapsed patient with a “standard” cytogenetic stage was a TN event and a non-relapsed patient with a “high” cytogenetic stage was an FP event. We show that the separation between the standard and high cytogenetic stage was realized according to the standard cytogenetic evaluation for separating patients with a high-risk mutation (poor prognosis in general) from patients without high-risk mutations [24,33].

### 3.3. Feature Ranking

To investigate which radiomics features mostly contribute to an effective stratification of the MM patients, we focused on the case of data set 3. The reason for this choice is because, when analyzed with HTF, this set provided, by far, the highest sensitivity values and, significantly, the highest CSI values among the three data sets considered. Therefore, we analyzed the feature compositions of the two axes produced by the application of PCA on the original feature space of this data set, made of 17 features. In Figure 5, we show the contribution of the 17 features to the first (light blue) and second (dark purple) principal component (PC). These contributions were weighted by the percentage of explained variance of the two PCs (77% and 9% for the first and second PC, respectively). A Mann–Whitney U-test on these features showed that just three of them did not pass the null hypothesis (*p* > 99%): “MaskMaximum”, which denotes the maximum grey level value in the mask segmenting the focal lesion (172.6 ± 64.4 in Cluster A; 321.9 ± 48.6 in Cluster B); “firstorderRange”, which denotes the range of the distribution of the voxel intensities (194.7 ± 61.8 in Cluster A; 343.4 ± 66.9 in Cluster B); and “ngtdmComplexity” (29.8 ± 24.9 in Cluster A; 79.4 ± 43.5 in Cluster B), which is a measure of the non-uniformity of the lesion image in the grey level intensity.

## 4. Discussion

This study demonstrates that AI supported radiomics realize a clustering of MM patients with a statistical reliability that, for some skill scores, is higher than the one provided by standard biochemical staging. The possibility to increase the predictive potential of the standard CT images of patients with multiple myeloma is clinically relevant for several reasons.

The first is that although MM is still considered a single disease, it is actually a collection of several different cytogenetically distinct plasma cell malignancies [2]. Trisomies and IgH translocations are considered primary cytogenetic abnormalities, and occur at the time of establishment of MGUS [2]. At the present time, there are three specific biomarkers for MM with an approximately 80% risk of progression to symptomatic end-organ damage in two or more independent studies: clonal bone marrow plasma cells ≥60%, serum free light chain (FLC) ratio ≥100 (provided involved FLC level is ≥100 mg/L), and more than one focal lesion on magnetic resonance imaging (MRI). It is known that almost all patients with MM eventually relapse and the choice of a treatment regimen at relapse is affected by many factors, including the timing of relapse, response to prior therapy, aggressiveness of relapse, and performance status (TRAP) [2]. Therefore, the prediction of relapse early is important to foresee a therapy. Second, several studies have correlated bone patterns in MM with their prognostic value using MRI and CT [9,10,17,18,21,34]. MRI can be used to differentiate up to five different patterns of plasma cell infiltration, including normal appearance, focal involvement, homogeneous diffuse infiltration, diffuse infiltration with additional focal lesions, and variegated or salt-and-pepper patterns; on the other hand, CT is well suited for small (below 5 mm) focal bone lesions due to its high spatial resolution capabilities [9].

The AI-based analysis of the radiomics properties extracted from the focal lesions essentially pointed out two aspects. First, the redundancy of the radiomics features seem to impact the prognostic power of the clustering methods. However, the stratification power increases when correlation-based and PCA-based reduction of redundancy processes are applied. Second, the use of a non-linear approach to clustering, namely HTF, seems to provide better results with respect to a more standard fuzzy clustering algorithm; this may be explained because of the high degree of heterogeneity that characterizes MM. The skill scores computed for each data set and each classification method helped us to determine which approach to redundancy reduction and which algorithm performs better for stratification purposes. Among the four skill scores, CSI probably represents the one that best interprets the outcomes of the confusion matrices in this context. Indeed, this score emphasizes the correct prediction of relapses in correspondence with a low rate of misclassification. Interestingly, the application of HTF on the focal features mostly correlating with the skeleton asset’s global properties (which are extracted by Bone-GUI) leads to the highest value for this score: this seems to point out a favorable prognostic role for the interplay between local and global descriptors of the MM bone tissue. In this case, the CSI value is higher than the discriminative value provided by the cytogenetic data, which supports the reliability of radiomics as a prognostic tool for MM clinical practice. This conclusion is confirmed by a bootstrap analysis performed on data set 3.

Data set 3 is made of the focal descriptors that mostly correlate with the whole skeleton’s asset properties. Therefore, this correlation analysis per se realizes a feature selection process whose outcome is a set of 17 features. A finer feature selection is provided by PCA, as shown in Figure 5. This figure and the related Mann–Whitney U-test point to a significant emphasis on properties related to the heterogeneity of the focal lesion, such as the Hounsfield unit range and maximum values found in the lesion, and the complexity, which measures the non-uniformity of the image and the presence of rapid changes in intensity.

We finally show that the data collection for this study has been realized by means of a single, specific CT scanner, so that the images we used for feature extraction were homogeneous. Recent studies [35] have shown that the characteristics of the extracted features may depend on non-tumor related factors like the signal-to-noise ratio of the experimental data. Therefore, in the case of studies that utilize data from more than one scanner, data homogenization should be implemented prior to the data extraction process [36].

## 5. Conclusions

This computational approach to the interpretation of radiomics focal features shows the potential for the stratification of relapsed and non-relapsed MM patients, and could represent a prognostic procedure for determining the disease follow-up and therapy. Concerning the technical issues to be discussed, the present study has several strengths: the use of clinically available CT images collected in the normal daily workup did not influenced patient care in any way. Second, we used a free open-source tool for radiomics assessment of the focal lytic lesions. Among the limitations of the present study, we acknowledge the retrospective nature, which did not allow for perfect timing between CT, diagnosis, and therapy or relapse. In addition, the evaluation of the radiomics features was made only with one open-source tool, and we did not evaluate whether the usage of other tools would have introduced variability to a significant extent. Finally, the overall number of patients included was relatively low: indeed, a correct sample size in radiomics is at least five times the number of extracted features [37], and this condition would require a population of at least 100 MM patients. Nonetheless, the possibility to obtain a cluster of features to identify relapses even in a 33 patient sample is in favor of the validity of this method. This initial study warrants prospective studies with a high number of patients, which are currently underway, in order to validate this approach, with the aim of implementing, it in a more systematic way, a method of obtaining a more robust prognostic score for MM patients.

Summing up the results of this study, we remind that our objective was to validate the feasibility of the automatic stratification of MM patients by means of an analysis of the descriptors extracted fromCT data within the framework of a radiomics retrospective study. This analysis showed that unsupervised AI can predict relapse within one year after transplantation and can identify a few imaging features associated with the heterogeneity of the focal lesion with a high prognostic value.

## Figures and Tables

**Figure 1 diagnostics-11-01759-f001:**
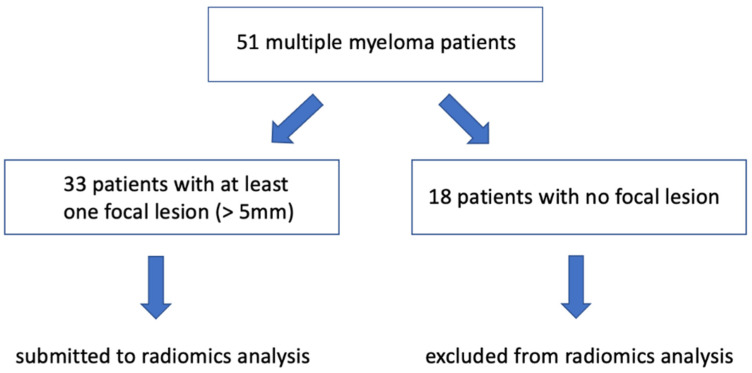
Flow diagram showing the initial number of participants in the study and those excluded because of not presenting focal lesions.

**Figure 2 diagnostics-11-01759-f002:**
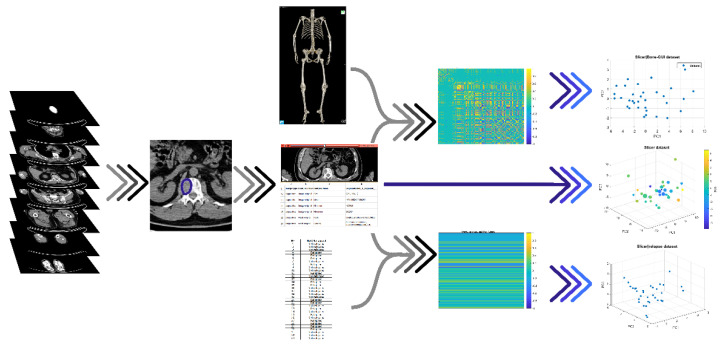
The pipeline of the radiomics features analysis. For each patient, the focal lesion was pointed out and the corresponding CT image was fed into a radiomics tool (Slicer), which computed 109 radiomics features; these descriptors were correlated with both the clinical outcome of the disease at one year, and the global radiological features extracted by means of a segmentation tool (Bone-GUI); the resulting mostly correlated features and the set of all local features were processed by means of two unsupervised AI algorithms (FCM and HTF) for stratification purposes.

**Figure 3 diagnostics-11-01759-f003:**
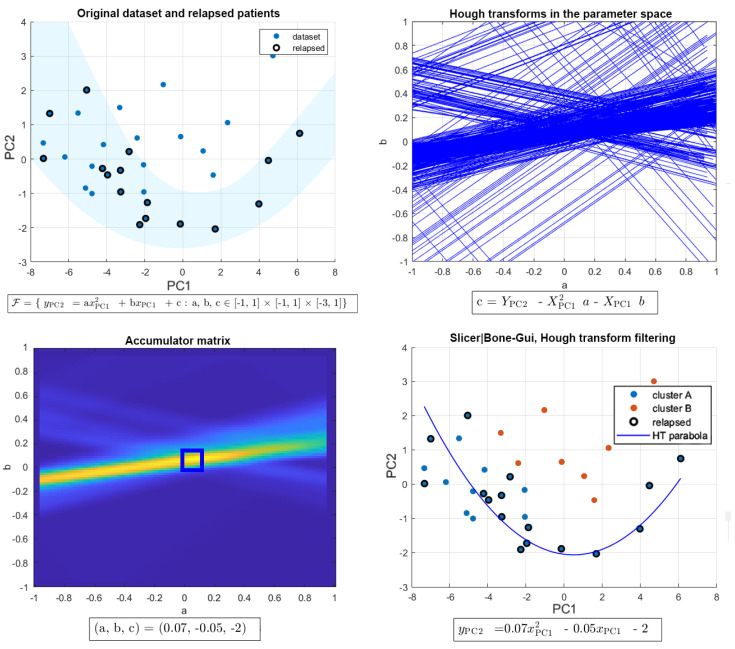
The HTF process for stratification. The feature space is constructed by applying PCA to the set of feature vectors (**top left panel**); for each point in the feature space the HT is computed with respect to the family of all parabolas (**top right panel**); the corresponding Hough accumulator is filtered by the smallest cube, including the cells with values higher than 50% of the accumulator maximum (**bottom left panel**); each filtered line is projected back into the feature space, thus generating the cluster of points associated to the parabola corresponding to the maximum of the Hough accumulator (**bottom right panel**).

**Figure 4 diagnostics-11-01759-f004:**
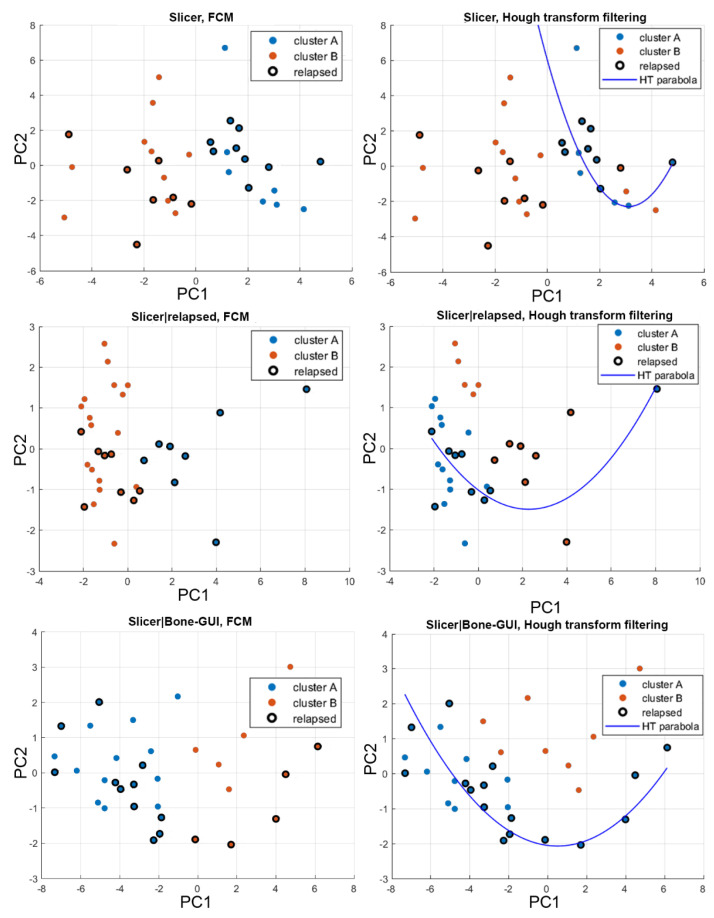
Clustering results for patients’ stratification. In all six panels, the clustering methods (FCM for the **left column** and HTF for the **right column**) are applied to the outcomes of the PCA analysis. The results are presented in two-dimensional spaces for easier reading. Cluster A (blue) contains the highest number of relapsed patients, the opposite is true for cluster B (orange); finally, black circles represent the patients that underwent relapse. Each row shows the results for a different data set: data set 1 is in top row, data set 2 is in middle row, and data set 3 in the bottom row.

**Figure 5 diagnostics-11-01759-f005:**
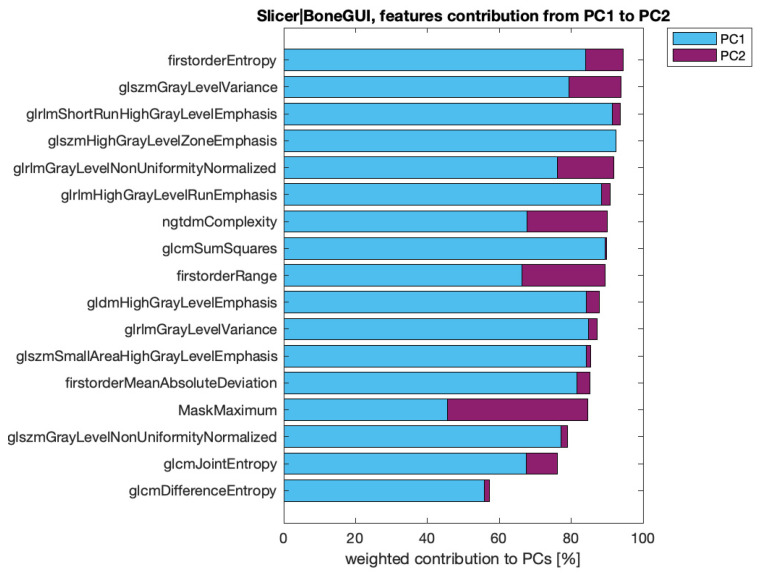
For data set 3, the first two principal components explain at least 80% of the data variance. In the figure, the make-up of the first two principal components is represented as follows: it is shown how much each feature contributes to the PC1 (light blue) and to the PC2 (dark purple), and these values are weighted for the percentage of data variation explained by each principal component considered.

**Table 1 diagnostics-11-01759-t001:** Clinical features of the 33 MM patients included in the analysis. R-ISS stage: I: ISS stage I and standard-risk CA by iFISH and normal LDH. II: Not R-ISS stage I or III; III: ISS stage III and either high-risk CA by iFISH or high LDH. CA—chromosomal abnormalities; iFISH—interphase fluorescent in situ hybridization; ISS—International Staging System; LDH—lactate dehydrogenase; MM—multiple myeloma; R-ISS—revised International Staging System.

Characteristic	Number	%
Patients	33	100
Age (years) Mean	56	
Age SD ^1^	6.7	
Males	21	66.4
Females	12	34.6
**Cytogenetics**		
Normal	22	66,7
High risk	11	33,3
Relapsed	17/33	51,5
Days before Relapse (mean)	1138	
Days of follow-up (mean)	1317	
**International Staging** **System**		
Stage I	15	45.4
Stage II	9	27.3
Stage III	9	27.3

^1^ Standard Deviation.

**Table 2 diagnostics-11-01759-t002:** Radiomics features extracted by means of image and correlation analysis.

Data Set Name	Vector Dimension	SW Tool	Feature Type	Correlation
Data set 1	109	Slicer	focal	no
Data set 2	8	Slicer	focal	relapses
Data set 3	17	Slicer	focal	global features

**Table 3 diagnostics-11-01759-t003:** Results of the clustering process provided by a fuzzy clustering method (FCM) and a non-linear filtering approach based on an extended version of the Hough transform (HTF). The symbol # denotes the cardinality of the set of vectors.

Method	Data Set	# of Vectors Cluster A	# of Vectors Cluster B	# of Relapses Cluster A	# of Relapses Cluster B
FCM	1	16	17	6	10
FCM	2	25	8	8	8
FCM	3	23	10	11	5
HTF	1	20	13	8	8
HTF	2	12	21	7	9
HTF	3	25	8	16	0

**Table 4 diagnostics-11-01759-t004:** Skill scores corresponding to the clustering analysis performed by means of FCM and HTF on the three data sets considered in the paper. The mean and standard deviation values are obtained by means of a bootstrap analysis that generated 100 random training sets made of 30 patients and, correspondingly, 100 random validation sets made of 13 patients. The last two rows contain the results of the analysis for the cytogenetics data associated with the patients.

Method	Data Set	Sensitivity	Specificity	Youden	CSI
FCM	1	0.46 ± 0.12	0.5 ± 0.14	−0.04 ± 0.13	0.3 ± 0.08
FCM	2	0.58 ± 0.35	0.55 ± 0.48	0.13 ± 0.15	0.3 ± 0.08
FCM	3	0.4 ± 0.24	0.55 ± 0.22	−0.06 ± 0.15	0.25 ± 0.12
HTF	1	0.38 ± 0.13	0.55 ± 0.16	−0.06 ± 0.15	0.25 ± 0.09
HTF	2	0.63 ± 0.19	0.33 ± 0.25	−0.04 ± 0.34	0.37 ± 0.16
HTF	3	0.87 ± 0.14	0.4 ± 0.13	0.27 ± 0.2	0.52 ± 0.1
Cytogenetics		0.45 ± 0.16	1.00 ± 0.02	0.44 ± 0.16	0.44 ± 0.16

## Data Availability

The datasets generated during and/or analyzed during the current study are available from the corresponding author on reasonable request.

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
