# Peer review of "Radiomics and Artificial Intelligence for Outcome Prediction in Multiple Myeloma Patients Undergoing Autologous Transplantation: A Feasibility Study with CT Data"

_diagnostics, 2021, doi:10.3390/diagnostics11101759_

Round 1

Reviewer 1 Report

The work presented is original in content and scientifically relevant.
The argument is well presented and methodologically adequate.

Author Response

Please find the response letter uploaded

Best

Michele Piana

Reviewer 2 Report

Some comments and questions:

  • It would be helpful to have a brief description of what the objective of the study is, in the introduction.
  • Figure 3 caption is not aligned with the figure. It is better to have the figure and caption together on the same page grouped together
  • Line 141 – Define PC1 and PC2
  • Why was datset 3 selected to investigate contribution of the radiomics featuresfor effective stratification of MM patients?
  • For figure 5, seems like it would be easier for interpretation to have % on the X-axis. Also, a label on x-axis could be helpful.
  • Looks like the data sets used in this study are taken from some instrument by some specific company. Would using a different instrument, say from a different company change anything? Assuming that there would be some difference in the quality of the data set or CT images depending on which instrument you use. How do you account for these differences?
  • One of the important pros of this study is that the techniques used are non-invasive and do not affect patient care in anyway and thus seems to have good potential. Also, the use of open source tools makes it easier to redo the study. However, the sample size is very low (33 patients) and makes you question the conclusion of this study and the reliability of the statistical deductions. But, overall this study can be a good starting point for further bigger and more systematic studies.

Author Response

1) It would be helpful to have a brief description of what the objective of the study is, in the
introduction

We agree with the referee that such description was missing in the first version of the manuscript.
The Introduction of the amended version now contains a brief statement that describes the
objective of this feasibility study

2) Figure 3 caption is not aligned with the figure. It is better to have the figure and caption together
on the same page grouped together

Done

3) Line 141 Define PC1 and PC2

This is now done in item 1 of the 'Clustering' subsection (current line 150)

4) Why was dataset 3 selected to investigate contribution of the radiomics features for effective
stratification of MM patients?

The referee is right: we should have explained this in the submitted version of the manuscript. The
reason of this choice is because this data set is associated to the highest CSI score values (when HTF
is applied) and therefore it is allegedly the one containing the largest amount of significant
information for stratification purpose. In the amended version of the paper, this is explained at the
beginning of the "Feature ranking" subsection of Section 3

5) For figure 5, seems like it would be easier for interpretation to have % on the X-axis. Also, a
label on x-axis could be helpful

Done

6) Looks like the data sets used in this study are taken from some instrument by some specific
company. Would using a different instrument, say from a different company change anything?

Assuming that there would be some difference in the quality of the data set or CT images
depending on which instrument you use. How do you account for these differences?

We agree with the referee that this issue is important. This study has been realized by means of a
single, specific instrument and so the data we used are homogenous. However, the quality of the
extracted features may depend on non-tumor related factors like, e.g., the signal-to-noise ratio of
the CT images. Therefore, in the case of studies performed by means of more than one scanner, a
data homogenization process should be implemented prior than the feature extraction step. We
have added a comment and two references on this in the amended version of the paper, at the end
of the Discussion section

7) One of the important pros of this study is that the techniques used are non-invasive and do not
affect patient care in anyway and thus seems to have good potential. Also, the use of open source
tools makes it easier to redo the study. However, the sample size is very low (33 patients) and
makes you question the conclusion of this study and the reliability of the statistical deductions.
But, overall this study can be a good starting point for further bigger and more systematic studies

We thank the referee for this comment. And we agree that the main limitation of this study is the
low size of the patients' sample. However, the aim of this first investigation was to assess the
feasibility of the AI-based approach to the stratification of MM patients. One of our current
objectives is to extend this study to a systematic analysis of a significantly higher population of
patients. In any case, we have added a brief comment on the sample size issue in the Conclusions
section of the amended version of the paper

Reviewer 3 Report

I commend the authors for using this approach and developing an AI tool for stratifying Myeloma patients based on CT data.

I find the presentation of some of the data is unclear, especially regarding Myeloma. How did they define standard vs high cytogenetics that they claim the tool outperformed it?

Why the authors used the ISS that no one uses anymore after the implementation of the R-ISS more than 8 years ago?

The sample number is small, and I don't see in methods how the authors tested if they are powered or if they did calculations for the optimal number of samples for the analysis.

Author Response

1) I commend the authors for using this approach and developing an AI tool for stratifying
Myeloma patients based on CT data

We thank the referee for this recommendation

2)
I find the presentation of some of the data is unclear, especially regarding Myeloma. How did
they define standard vs high cytogenetics that they claim the tool outperformed it?

We have defined the standard vs high cytogenetics separation according to the standard cytogenetic
evaluation for separating patients with high-risk mutation (poor-prognosis in general) from patients
without high-risk mutations. The reference is: Roche-Lestienne C, Boudry-Labis E, Mozziconacci MJ.
Cytogenetics in the management of "chronic myeloid leukemia": an update by the Groupe
francophone de cytogénétique hématologique (GFCH). Ann Biol Clin (Paris). 2016 Oct 1;74(5):511-
515. English. doi: 10.1684/abc.2016.1151. PMID: 27477825. In brief: High risk: presence of del(17p)
and/or translocation t(4;14) and/or translocation t(14;16): Standard risk No high-risk chromosomal
abnormalities (J Clin Oncol. 2015 Sep 10; 33(26): 28632869. doi: 10.1200/JCO.2015.61.2267). We
have specified this in the text of the revised version of the manuscript, at the end of the "AI-based
analysis" subsection, and we have added the corresponding reference

3) Why the authors used the ISS that no one uses anymore after the implementation of the R-ISS
more than 8 years ago?

We apologize for this inaccuracy: we have actually used the R-ISS and we agree this is the standard.
We have corrected and specified this in the amended version of the manuscript and we have added
a reference. Further, the caption of Table 1 has been updated accordingly

4)
The sample number is small, and I don't see in methods how the authors tested if they are
powered or if they did calculations for the optimal number of samples for the analysis.

We agree with the referee: a low sample size is the main drawback of this study. Since in radiomics
a reliable criterion is that the number of subjects is at least five times the number of variables (see,
e.g., Sollini et al, EJNMMI, 46, 2656, 2019) and since in our analysis we considered around twenty
features per patients, the population considered for this kind of investigation should be of around
100 subjects. However, this is a first feasibility study and we are currently setting up a radiomics
investigation of MM that includes a population characterized by the correct sample size. We have
added a comment on this in the conclusion section of the manuscript

This manuscript is a resubmission of an earlier submission. The following is a list of the peer review reports and author responses from that submission.

Round 1

Reviewer 1 Report

The authors suggested that radiomics and AI-based computational analysis of CT data was prognostic in patients with multiple myeloma who underwent stem cell transplantation. The application of AI-based analysis to predict the prognosis of multiple myeloma is considered as a new and important study. But, this study has some major limitations to be published in this journal.

1)  The patient’s clinical information such as frontline therapy, cytogenetic risk stratification, or maintenance therapy after transplantation, were not sufficiently provided in this study, and we don’t know whether the high-risk and standard risk group were properly classified.

2) Recently, PET/CT or MRI image has been mainly used to analyze the prognosis of multiple myeloma. The authors did not fully explain why CT images were chose rather than PET or MRI in this study.  

Reviewer 2 Report

cancers-1286833

Radiomics and Artificial Intelligence for Outcome Prediction in Multiple Myeloma Patients Undergoing Autologous Transplantation: a Feasibility Study with CT Data

The article " Radiomics and Artificial Intelligence for Outcome Prediction in Multiple Myeloma Patients Undergoing Autologous Transplantation: a Feasibility Study with CT Data" (cancers-1286833) by Schenone D, et al. demonstrated that radiomics using CT data predicted possibility of relapse among the myeloma patients received autologous stem cell transplantation (ASCT). Radiomics using CT is a new prognostic approach, and might less invasive and useful. On the other hand, several analyses about radiomics using PET/CT or MRI among the myeloma patients has already done. Therefore, the novelty of this study might be poor. Besides, there are several major and minor issues to be addressed as below.

Major issues

  1. The author should add the data about patient characteristics including imaging findings (diffuse or local, and bone or extramedullary diseases), induction treatment, and treatment response after ASCT.

  1. The author should analyze the survival time (progression free survival) using the Kaplan-Meier method.

Minor issues

  1. I considered that Table 2 was not needed because the author did not describe the difference between data set 2 and 3. The predictive value of data set 3 seemed to be superior to those of data set 3. When my understanding was correct, the author could add the difference between data set 2 and 3 if possible.

  1. The author could add the imaging findings in the relapsed patients in results or discussion.

  1. The author could add statistic correlation between radiomics data and chromosomal abnormalities.

  1. The author could describe the difference of radiomics using CT compared with PET/CT and MRI if possible.

  1. In your study, the sensitivity of radiomics using CT was superior to those of high-risk cytogenetic abnormalities while the specificity of radiomics using CT was inferior to those of high-risk cytogenetic abnormalities. The author could discuss these results.